# NADH-GOGAT Overexpression Does Not Improve Maize (*Zea mays* L.) Performance Even When Pyramiding with NAD-IDH, GDH and GS

**DOI:** 10.3390/plants9020130

**Published:** 2020-01-21

**Authors:** Rafael A. Cañas, Zhazira Yesbergenova-Cuny, Léo Belanger, Jacques Rouster, Lenaïg Brulé, Françoise Gilard, Isabelle Quilleré, Christophe Sallaud, Bertrand Hirel

**Affiliations:** 1Institut Jean-Pierre Bourgin, Institut National de la Recherche Agronomique (INRA), Centre de Versailles-Grignon, Unité Mixte de Recherche 1318 INRA-AgroParisTech, RD10, 78026 Versailles, CEDEX, Francecunyzhaz@gmail.com (Z.Y.-C.); bellangerleo@gmail.com (L.B.); lenaig.brule@inra.fr (L.B.); isabelle.quillere@inra.fr (I.Q.); 2Departamento de Biología Molecular y Bioquímica, Facultad de Ciencias, Universidad de Málaga, Campus de Teatinos, 29071 Málaga, Spain; 3BIOGEMMA, GM Trait Discovery, Site de la Garenne, Route d’Ennezat, CS 90126, F-63720 Chappes, France; jacques.rouster@biogemma.com (J.R.); christophe.sallaud@biogemma.com (C.S.); 4IPS2/Plateforme Métabolisme-Métabolome, Institute of Plant Sciences Paris-Saclay, Centre National de la Recherche Scientifique (CNRS), Institut National de la Recherche Agronomique (INRA), Université Paris-Sud, Université Evry, Université Paris-Diderot, Université Paris-Saclay, 91190 Gif-sur-Yvette, France; Francoise.Gilard@Ips2.Universite-Paris-Saclay.Fr

**Keywords:** amino acids, biomass, carbon, genetic manipulation, kernel yield, maize, nitrogen, organic acids

## Abstract

Maize plants overexpressing NADH-GOGAT were produced in order to determine if boosting 2-Oxoglurate production used as a carbon skeleton for the biosynthesis of amino acids will improve plant biomass and kernel production. The NADH-GOGAT enzyme recycles glutamate and incorporates carbon skeletons into the ammonium assimilation pathway using the organic acid 2-Oxoglutarate as a substrate. Gene pyramiding was then conducted with NAD-IDH and NADH-GDH, two enzymes also involved in the synthesis of 2-Oxoglurate. NADH-GOGAT overexpression was detrimental for shoot biomass production but did not markedly affect kernel yield. Additional NAD-IDH and NADH-GDH activity did not improve plant performance. A decrease in kernel production was observed when NADH-GDH was pyramided to NADH-GOGAT and NAD-IDH. This decrease could not be restored even when additional cytosolic GS activity was present in the plants overexpressing the three enzymes producing 2-Oxoglutarate. Detailed leaf metabolic profiling of the different transgenic plants revealed that the NADH-GOGAT over-expressors were characterized by an accumulation of amino acids derived from glutamate and a decrease in the amount of carbohydrates further used to provide carbon skeletons for its synthesis. The study suggests that 2-Oxoglutarate synthesis is a key element acting at the interface of carbohydrate and amino acid metabolism and that its accumulation induces an imbalance of primary carbon and nitrogen metabolism that is detrimental for maize productivity.

## 1. Introduction

Large quantities of nitrogenous fertilizers are required to attain maximal yields in cereals such as maize, wheat, and rice, which account for 70% of the worldwide food production [1,2]. In consideration of both environmental and economic concerns, several studies were thus conducted to improve nitrogen use efficiency (NUE) in these crops [3,4]. The efficiency of N use is generally defined as the yield obtained per unit of available N in the soil (supplied by the soil + N fertilizer). It is the product of absorption efficiency (amount of absorbed N/quantity of available N) and use efficiency (yield/absorbed N), [4,5].

Thus, studies in which physiological and molecular genetic approaches were combined have been undertaken in order to identify the limiting steps involved in the control of N uptake, assimilation and recycling in many crops [3,4] including maize [6]. Different approaches, including whole-plant physiological [7] and quantitative genetic studies [8] were originally developed to identify both structural and regulatory candidate genes, as well as the key enzymatic reactions involved in the control of NUE. Such combined approaches have formulated the hypothesis that a number of steps spanning from N uptake to N incorporation into organic molecules such as amino acids could be involved in the control of maize productivity.

Among these steps, the reaction catalysed by the enzyme glutamine synthetase (GS; EC 6.3.1.2) was intensively studied. This was largely due to its pivotal role in the process leading to the incorporation of inorganic N into organic molecules and its ubiquitous function in different parts of the plant through the activity of various isoenzymes located in diverse organs and cellular compartments [9,10]. Whether derived from nitrate, ammonium ions, N fixation, or generated by other reactions within the plant that release ammonium, most, if not all the N in a plant is channeled through the reactions catalyzed by the various cytosolic and plastidic GS isoenzymes. Ammonia, which is the ultimate form of inorganic N available to the plant, is then incorporated into an organic molecule through the reaction catalyzed by the GS enzyme to synthesize glutamine [11,12]. Improvement in NUE increased biomass and higher grain yield was obtained in several species such as rice, maize, barley, pea and poplar, transgenic plants that overexpressed GS1 or GS2, [13,14]. One of the most remarkable results was the increase in grain yield obtained in maize plants overexpressing GS1, thus leading to the hypothesis that GS is one of the key drivers in plant productivity, at least with regards to the efficiency of N assimilation [15].

The second enzyme involved in ammonium assimilation, which was proposed as being a good candidate for improving N assimilation efficiency, is glutamate synthase, also called glutamine:2-Oxoglutarate aminotransferase (GOGAT). The reaction catalyzes the transfer of the amide amino group of glutamine to 2-Oxoglutarate allowing the synthesis of two molecules of glutamate. GOGAT recycles glutamate and incorporates carbon (C) skeletons into the GS/GOGAT cycle using the organic acid 2-Oxoglutarate as a substrate. In this coupled reaction, 2-Oxoglutarate originates from photosynthetic CO_2_ assimilation, via the glycolysis pathway and the Krebs cycle [16] at the interface between C and N metabolism. The GOGAT enzyme in plants is present in two distinct forms. One uses reduced ferredoxin (Fd) as the electron donor (EC 1.4.7.1) in photosynthetic tissues. The second enzyme used reduced nicotinamide adenine dinucleotide (NADH) as the electron donor in the reaction catalyzed by GOGAT occurring predominantly in non-photosynthesizing cells (EC 1.4.1.14). As also identified for the different GS isoenzymes, the different genes encoding GOGAT have specific and non-redundant roles in the different pathways of N metabolism. For example, in rice, one of the two NADH-GOGAT forms (NADH-GOGAT1) is important in the primary assimilation of ammonium in roots, whereas the other form (NADH-GOGAT2) plays a role in the remobilization of N during leaf senescence. Although knock-out mutations of NADH-GOGAT2 genes have been shown to be detrimental for seed production, it is still not fully clear whether increasing the enzyme activity will be beneficial in terms of plant productivity [17]. Nevertheless, there are strong lines of evidence based on genetic studies performed on wheat that NADH-GOGAT could be a key player in the control of grain quality [18].

The source of C for ammonium assimilation is the organic acid 2-Oxoglutarate, which can be synthesized either in the mitochondria or in the chloroplast via the activity of the NAD-dependent (EC 1.1.1.41) or the NADP-dependent (EC 1.1.1.42) form of isocitrate dehydrogenase (IDH) enzymes respectively [19]. However, it is generally admitted that the conversion of isocitrate to 2-Oxoglutatrate in the mitochondria is regarded as being one of the main sources of C that can be used by the GS/GOGAT cycle [20]. Using Arabidopsis mutants deficient in the expression of three genes encoding NAD-IDH, it has been shown that the enzyme does not limit N assimilation, although a mild reduction of its activity led to a noticeable alteration in nitrate assimilation in transgenic tomato plants [19]. To date, the impact of an overexpression of IDH has not been studied, although it has been proposed that the tricarboxylic acid cycle in plants could be a target to improve plant productivity not only due to its ability to produce energy, but also to provide C skeletons for the nitrogen assimilation pathway [21].

Another enzyme that is able to synthesize 2-Oxoglutarate catalyzes is the enzyme (NADH-GDH; E.C. 1.4.1.2), which catalyzes a reaction involving glutamate and ammonium. As such, the enzyme has the potential capacity to assimilate inorganic N by combining ammonium with 2-Oxoglutarate to form glutamate. GDH also has the capacity to deaminate glutamate by means of a reversible reaction. However, most studies have shown that GDH operates in the reverse direction of glutamate deamination to release organic acids when the cell is C-limited [22], even when the activity of the enzyme is markedly increased in genetically modified plants [23]. In most plant species, two distinct genes encoding α and β GDH subunits have been identified to date. These two distinct subunits are able to assemble apparently at random into active hexamers. The relative proportion of the α- and β-subunits, and hence, the isoenzyme pattern observed following electrophoresis, varies with plant organ, N source and light regime [24]. The physiological significance of such variability in the GDH subunit composition is still not clearly understood. However, there are strong lines of evidence that each subunit plays a specific function at the interface of C and N metabolism by controlling glutamate homeostasis and thus, the subsequent synthesis and export of derived amino acids [23]. In several studies, overexpressing a NADPH-GDH gene originating from fungi or unicellular organisms led to an increase in ammonium assimilation accompanied by an improvement of crop agronomic performances or stress resistance [25,26]. In contrast, the overexpression of a NADH-dependent enzyme from plants had a negative impact on growth and development [23].

To study the role of the different enzymes involved in providing C to the ammonium assimilation pathway and in the subsequent synthesis of amino acids, a large number of reverse and forward genetic approaches have been conducted over the past two decades both in model and crop species [4,21]. Both glutamine and glutamate are two key N metabolites that act as amino group donors to form other amino acids used for transport, structural and storage protein synthesis and the synthesis of nucleotides for the formation of RNA and DNA [11,27]. In addition, these two amino acids are used as transport molecules to provide organic N to developing and storage organs [28]. Thus, the hypothesis that enhancing glutamate and glutamine production via an increase in the flux of C and N molecules used for their synthesis could be beneficial for plant productivity has been put forward on a regular basis. Although modulating the level of expression of the cognate genes gave information on their role when considered individually, it was often proposed that more work would be required to determine if altering part or the entire ammonium assimilation pathway could be more efficient to improve both NUE and plant performance [13].

In the present study, an attempt was thus made to overexpress NADH-GOGAT and stacked with NAD-IDH, NADH-GDH1 and cytosolic GS (GS1.3) in maize. These genetic manipulations were conducted in order to determine if boosting the supply of C skeletons and the production of glutamate and glutamate could be beneficial in terms of plant productivity in an economically important crop such as maize [3,29]. Detailed leaf metabolite profiling was conducted at two important stages of plant development, including the vegetative and the grain filing periods, in order to determine if there was any relationship between shoot biomass, kernel production and primary and secondary metabolite accumulation. Whether the pattern of leaf metabolite accumulation resulting from the overexpression of the enzymes involved in the ammonium assimilation pathway can be used as a biochemical marker to select maize lines exhibiting improved agronomic performance is discussed.

## 2. Results

### 2.1. Overexpression of NADH-GOGAT and Gene Pyramiding with NAD-IDH, NADH-GDH1 and GS1.3

To produce lines containing elevated amounts of NADH-GOGAT alone and stacked with NAD-IDH, NAD-IDH + NADH-GDH1 and NAD-IDH + NADH-GDH1 + GS1.3, four different constructs composed of the full length cDNAs encoding the four enzymes were fused to different promoters in order to direct their expression constitutively in shoots (Appendix A). To produce maize lines containing elevated amounts of GS1 in the leaves both in the mesophyll and in the bundle sheath cells [15], the *Gln1-3* full length cDNA, was fused to the cassava vein mosaic virus promoter (*CsVMV* promoter) and to the promoter of the maize Rubisco small subunit (*RbcS*). These four different constructs named T02291, T02289, T02308 and T02312, respectively were introduced into a recombinant plasmid (pBIOS01957) used for plant transformation via *Agrobacterium tumefaciens.*

Five or six independent transformants were produced for each of the four constructs. After an initial cross of the primary transformant (T_0_ plant) with the pollen of the wild type (WT, A188), two rounds of self-pollination were performed in order to obtain plants for which the cob carried only homozygous seeds.

A RT-qPCR analysis was then conducted to quantify the level of expression of the NADH-GOGAT, NAD-IDH, NADH-GDH1 and GS1.3 transgenes in leaves of the transgenic maize plants harvested at the V stage. This analysis showed that for each of the four constructs (T02291, T02289, T02308, T02312), there was an accumulation of NADH-GOGAT, NAD-IDH, NADH-GDH1 and GS1.3 transcripts in comparison to the untransformed WT plants. Depending on the transgene, the number of transgenes and on the transformation event, the level of expression of the four transgenes was variable with up to 14-fold variation from one transgenic event to the other. However, it can be seen that for each of the four constructs, at least four independent transgenic events exhibited a similar pattern of transgene expression. On the basis of the transgene expression pattern for each construct, four independent transgenic events were selected for further metabolic phenotyping and for the measurement of enzyme activities (Appendix A). Shoot biomass and yield-related traits were measured using all the transgenic events produced for each construct.

Figure 1 shows the level of leaf enzyme activities measured in the different transgenic plants overexpressing NADH-GOGAT, NAD-IDH, NADH-GDH1 and GS1. At least a 3-fold increase in the different enzyme activities was observed in comparison to A188, the untransformed control line. For the two constructs (T02289 and T02312), only a 2-fold increase in NADH-GOGAT activity was observed. In roots, NADH-GOGAT activity was practically unchanged and not statistically different compared to that of the untransformed control lines (data not shown). For NADH-GDH, the activity corresponds to the aminating reaction. Similar differences were observed when the deaminating enzyme was measured (data not shown).

### 2.2. Phenotypic Characterization of Transgenic Plants

The five to six independent transformation events from each homozygous T_2_ transgenic line overexpressing NADH-GOGAT (line T02291), NADH-GOGAT/NAD-IDH (line T02289), NADH-GOGAT/NAD-IDH/NADH-GDH1 (line T02308), NADH-GOGAT/NAD-IDH/NADH-GDH1/GS1.3 (line T02312), (Appendix A) and untransformed WT plants (line A188) were grown until maturity on a complete nutrient solution containing optimal amounts of N in the form of 8 mM NO_3_^−^ + 2 mM NH_4_^+^. Plant biomass production expressed on a total dry weight (DW) basis of the shoots of lines T02291, 702289 and T02312 was approximately 25% lower than the WT. Only in line T02308 shoot DW was not significantly different compared to the three other transgenic lines and the WT (Figure 2A).

Kernel yield in transgenic lines T02291, T02289, T0312 was not significantly reduced compared to the WT. Compared to the WT, a 50% decrease in KY was observed in line T022308 (Figure 2B). No differences were observed when comparing TKW of the different transgenic lines and of the WT, thus indicating that kernel size was not significantly modified in the four different transgenic lines (Figure 2C). Changes in KN between the different transgenic lines and the WT were similar to that observed for KY, thus indicating that yield was reduced because these lines produced less kernels while maintaining their weight (Figure 2D).

### 2.3. Leaf Metabolite Profiles at Two Stages of Plant Development

Gas Chromatography coupled to Mass Spectrometry (GC/MS) analyses of the leaf metabolome were carried out on the four selected independent transgenic events for each of the four different transgenic maize lines T02291, T02289, T0312 and T022308 that had been grown in the glasshouse (Appendix A). In the leaf samples taken at the vegetative stage of plant development (V) and during the grain filling period fifteen days after silking (15DAS), 128 water-soluble leaf metabolites were detected (Appendix A). Following normalization of the amounts of metabolites (Appendix A), a hierarchical clustering analysis (HCA) showed that in the different transgenic lines and in the WT, metabolites displayed a clear difference in the level of accumulation between the V stage and 15DAS. About half of the 128 detected metabolites were more abundant at the V stage while for the other half, they were present in higher amounts 15DAS (Appendix A). A Principal Component Analysis (PCA) was then performed to obtain a visual representation of the differences in the metabolic profile of the WT and of the four different transgenic maize lines T02291, T02289, T0312 and T022308. Leaf metabolites were separated along axis 1 (PC1) between the V stage and 15DAS explaining 50.4% variation in their profile (see also Appendix A). Along axis 2 (PC2), it can be seen that only at the V stage were the WT plants separated from the transgenic lines, all grouped in a different cluster. Axis 2 explains the limited 8.6% variation observed for the leaf metabolite profile between the WT and the four different transgenic lines occurring mainly at the V stage (Figure 3). Therefore, only the leaf metabolite content at the V stage was used for further comparison of the WT and transgenic lines and for the corresponding statistical analyses.

After *t*-test statistical analysis (*p* ≤ 0.05) followed by a FDR test and correction, only 51 metabolites were found to be significantly different between the WT and the four different transgenic lines at the V stage (Appendix A). Most of the metabolites were amino acids, organic acids of the TCA cycle and carbohydrates. At 15DAS, there were very few metabolites exhibiting a decrease or an increase in the different transgenic lines, thus confirming the results obtained following PCA analysis (Figure 3). In Table 1, are presented the most important changes in the relative content of metabolites between the WT (A188) and the transgenic plants. An increase in the amount of 2-Oxoglutarate was only observed in the plant overexpressing NADH-GOGAT and GOGAT + NAD-IDH. In all the transgenic lines, a decrease of carbohydrates involved in cell wall biosynthesis, such as arabinose and xylose, was detected. Such a decrease was also observed for the two rare hexoses, tagatose and pscicose. The two intermediates of the glycolytic pathway glucose-6P and fructose-6P were present in lower amounts only in the transgenic plants that did not express GS1.3. In the NADH-GOGAT overexpressors there was an increase in the amount of several amino acids, notably those derived from glutamate, such as alanine, aspartate and proline. Following gene pyramiding with NAD-IDH, the increase in aspartate did not occur anymore. The increase in proline and alanine observed in the NADH-GOGAT over-expressors did not occur when there was an additional GDH and GS1 activity, respectively.

### 2.4. Correlations between Leaf Metabolites, Shoot Biomass and Yield-Related Traits

In order to identify possible functional relationships between leaf metabolite accumulation at the V stage and shoot biomass and kernel yield-related traits, their Pearson correlation coefficients were calculated using the entire metabolomic dataset obtained with the four transgenic lines and the WT. Pearson correlations were then calculated to establish possible relationships between the leaf metabolite content, shoot dry weight and yield-related traits (GY, KN and TKW). They are shown in Appendix A (Pearson correlations) together with their *p*-value (≤0.05). Hierarchical clustering of the Pearson correlations allowed the generation of six modules of co-regulated metabolite accumulation (Appendix A). In the subsequent analyses, each of these six modules were represented by a different colour (red, blue, orange, turquoise, yellow and pink). For example, one of the main characteristics of the yellow module that exhibited a correlation with the agronomic traits was the presence of carbohydrates and a few organic acids. The blue module contained mostly amino acids and a number of C-containing molecules. The orange module was mainly composed of organic acids including those belonging to the TCA cycle. (Appendix A). The correlations between metabolites and between shoot biomass and KY, for which the most interesting correlations were obtained, are visualized in the network diagram presented in Figure 4A,B respectively. Their values and the corresponding statistical analyses are presented in Appendix A.

The most interesting results were the identification of positive of correlations between shoot biomass and a number of carbohydrates, such as arabinose (0.59), psicose (0.65) tagatose (0.65) and xylose (0.56), as well as secondary metabolites such as quinic acid (0.63) and shikimic acid (0.62), all of which belonged to the yellow module. In contrast, negative correlations were found with metabolites belonging to the blue module including amino acids such as isoleucine (−0.45), leucine (−0.55), proline (−0.46), threonine (−0.46) and tryptophan (−0.63), the organic acid *trans*-aconitate (−0.46) and the vitamin-tocopherol (−0.49). Negative correlations were also found between shoot biomass fatty acids such as oleic acid (−0.63), linoleic acid (−0.63), palmitic acid (−0.59) and stearic acid (−047), all of these molecules being grouped in the orange module.

As for shoot biomass, positive correlations were obtained between the amount of arabinose (0.46), psicose (0.46), tagatose (0.44), xylose (0.46), quinic acid (0.49), shikimic acid (0.51) and KY. In addition, glucose-6-phosphate (0.46) was also positively correlated with KY. The negative correlations obtained between KY with isocitrate (−0.60) and citrate (−0.60), three organic acids belonging the TCA cycle and fucose (−0.47), were different compared to those obtained with shoot biomass.

## 3. Discussion

When the expression of key enzymes involved in 2-Oxoglutarate biosynthesis (NAD-IDH and NADH-GDH) and ammonium assimilation (NADH-GOGAT and GS1) was enhanced, changes in the leaf metabolite profile occurred almost exclusively at the vegetative stage. Such a finding suggest that, at least in maize, their role is likely to be predominant during the N assimilation phase and that other enzymes involved in different pathways, or other differentially expressed isoenzymes are involved in N redistribution and recycling during the grain filling period [7]. Such a hypothesis is illustrated by the fact that in the different transgenic lines overexpressing NADH-GOGAT alone, or pyramided with NAD-IDH, NADH-GDH and GS1, there was an inversion in the amount of the two main classes of metabolite profiles when the leaf metabolic profile at the V stage and 15DAS were compared (Appendix A).

There are only a limited number of reports describing the overexpression of GOGAT in plants using either the pyridine nucleotide-dependent form [30,31], or the ferredoxin-dependent enzyme [32]. In these studies, a 50% to 80% increase in enzyme activity was observed, irrespective of whether the native enzyme or an enzyme originating from another species was used for the genetic manipulation. In the maize plants overexpressing NADH-GOGAT, on average, a 2- to 8-fold increase in NADH-GOGAT activity was obtained in the transgenic plants T02291, T02289, T02308 and T02312, in which a heterologous gene from wheat fused to a constitutive promoter was introduced. The wheat NADH-GOGAT cDNA was used because it allowed to increase the enzyme activity in transgenic cereals such as wheat and maize. Such differences in the level of enzyme activity from one construct to another can be explained by the large variability often observed in the level of expression of transgenes due to random insertion into the genome and to the presence of unknown regulatory mechanisms acting either at the transcriptional or post-translational levels [33,34]. Moreover, the NADH-GOGAT is a large gene of more than 13 kb, which can induce unpredicted side effects in the neighboring genome coding sequences.

Overexpression of NADH-GOGAT activity was detrimental only for shoot biomass production, irrespective of the increase in enzyme activity in the four types of transgenic plants. Such a negative impact of NADH-GOGAT overexpression has already been observed in the progeny of transgenic rice [31]. Only in tobacco plants overexpressing constitutively the enzyme from alfalfa was an increase in shoot dry weight observed, and then, only at the vegetative stage. As for several genetic manipulations involving enzymes of the N assimilation pathway, contrasting results were obtained which is likely due to the nature of transgene and of the species or genetic background used for plant transformation [13].

In our study, the negative impact of NADH-GOGAT overexpression can be explained by the fact that an increase in enzyme activity leads to the production of an excess of glutamate in the leaves. However, glutamate does not accumulate, probably because it is readily used for the synthesis of derived amino acids such as alanine, leucine, isoleucine aspartate, serine, tryptophan and proline and aspartate-derived amino acids except methionine [35] (Figure 5). Such a hypothesis is supported by the finding that all these amino acids derived from glutamate are present in higher amounts in the NADH-GOGAT overexpressors. Consequently, 2-Oxoglutarate accumulates probably because it is not used as a C skeleton for the synthesis of glutamine via the GS/GOGAT pathway, thus explaining why there was no accumulation of glutamine, which also uses glutamate as a substrate. As C-containing molecules are necessary for the synthesis of alanine, aspartate, valine, leucine, isoleucine and proline, notably those derived from pyruvate, we observed a decrease in the amount of several carbohydrates (Figure 5). These carbohydrates originate from sucrose degradation such as glucose-6-P and fructose-6-P, or from those used for cell wall biosynthesis, such as arabinose and xylose [36,37] or to synthesize tagatose and psicose rare sugars involved in metabolic signaling [38]. The carbohydrates that originate from sucrose degradation are used to feed glycolysis and the TCA cycle to produce C skeletons for the amino acid synthesis. It is likely that such a compensatory mechanism is a way to regulate the balance between C and N assimilation while maintaining, at the same time, glutamate homeostasis. Glutamate plays a central role in plant nitrogen metabolism, not only as a precursor for the synthesis of a wide range of N-containing compounds, but also as a key signaling molecule at the interface of C and N metabolism [27]. As such, there is evidence that under most circumstances, the plant is able to maintain the soluble concentration within fairly narrow limits, which could explain the metabolic regulation occurring in the transgenic maize plants overexpressing NADH-GOGAT.

In the three different types of transgenic lines (T02289, T02308 and T02312) in which NAD-IDH was overexpressed, the increase in the enzyme activity approximately 4-fold compared to the untransformed control plants. The presence of additional NAD-IDH activity in the NADH-GOGAT over-expressors, did not modify either plant biomass production or kernel weight in comparison to the transgenic plants overexpressing NADH-GOGAT alone (T0291). The sorghum NADH-IDH cDNA was used because it allowed to increase the enzyme activity in transgenic cereals such as wheat and maize. The leaf metabolite profile of the plants overexpressing NADH-GOGAT and NAD-IDH was almost identical compared to that of the NADH-GOGAT over-expressors, which could explain why pyramiding the gene encoding NAD-IDH did not induce changes in plant performance. Our finding agrees with the work of Lemaître et al. [20], in which they showed, using a reverse genetic approach, that in Arabidopsis, NAD-IDH did not limit N assimilation and did not have any impact on carbohydrate metabolism. It is also likely that the increase in 2-Oxoglutarate observed in plants overexpressing NADH-GOGAT was not enhanced further when another enzyme producing the organic acid, such as NAD-IDH, is overexpressed simultaneously. However, further work will be required to ascertain such a hypothesis by overexpressing only NAD-IDH in maize or using mutants deficient in the enzyme activity.

In line with the hypothesis that increasing the production of 2-Oxoglutarate would boost N assimilation, increasing the activity of NADH-GDH, another enzyme with the capacity to synthesize 2-Oxoglutartate [39], was achieved by pyramiding the corresponding gene to the NADH-GOGAT/NAD-IDH over-expressors. In the two types of transgenic plants overexpressing NADH-GDH1 (T02308 and T02312), the increase in the enzyme activity was, on average, around 3-fold compared to the control line A188). In the NADH-GOGAT/NAD-IDH/NADH-GDH1 overexpressors, the amounts of 2-Oxoglutarate, Aspartate, Proline, Tryptophan and Lysine were again comparable to that of the untransformed control plants as was plant biomass production. In contrast, KY was reduced by about 50% compared to the untransformed control plants. In tobacco, a comparable increase in NADH-GDH activity obtained by overexpressing the gene from *Nicotiana plumbaginifolia* induced a reduction in plant biomass production, likely because the use of glutamate for amino acid biosynthesis was strongly perturbed [23]. Interestingly in the maize NADH-GOGAT/IDH over-expressors exhibiting additional GDH activity, there was no increase in 2-Oxoglutarate in the transgenic plants overexpressing NADH-GOGAT alone (T02291), or NADH-GOGAT and NAD-IDH (T02289). These unexpected findings suggest that 2-Oxoglutarate could be used to assimilate ammonium in order to restore the homeostasis of glutamate and maintain the level of the organic acid to a physiological level comparable to that of the untransformed control plants. It would thus be interesting to verify experimentally this hypothesis by feeding the leaves of the GOGAT/IDH and GDH over-expressors with ^15^N-ammonium [40] in order to verify if they are able to synthesize ^15^N-glutamate.

In order to determine if the synthesis of glutamine could be boosted when the synthesis of glutamate and 2-Oxoglutarate was also enhanced, a fourth gene encoding Cytosolic GS (GS1) under the control of two promoters allowing the overexpression of the gene in the leaf mesophyll and in the bundle sheath cells was stacked to the cassette of genes, allowing the overexpression of NADH-GOGAT, NAD-IDH and NADH-GDH1. In the transgenic plants (T02312) overexpressing the four enzyme activities simultaneously, we did not observe an accumulation of amino acids as in the three other transgenic lines T02291, T02289 and T02308. Such a finding suggests that the overexpression of GS1 counteracts the diversion of glutamate used for the synthesis of the amino acids derived from Pyruvate and that are able to use glutamate as a substrate. However, this shift in the flux of organic N does not have any major impact on the utilization of carbohydrates, which are probably still necessary to provide C skeletons to the GS/GOGAT pathway. Although it was only a trend, the impact of this shift in the amino acid biosynthetic pathway was to partially restore the detrimental impact of NADH-GDH overexpression on kernel production. This result is in line with a previous study in which the key role of GS1 in kernel production in maize was demonstrated [15].

## 4. Material and Methods

### 4.1. Production of Maize Transgenic Lines

Transgenic maize A188 lines were produced in which four different constructs were introduced to overexpress, in a constitutive manner, NADH-GOGAT alone (construct T02291) or stacked with NAD-IDH (construct T02289), NAD-IDH + GDH1 (construct T02308) and NAD-IDH + GDH1 + GS1.3 (construct T02312). Details of the four different constructs are presented in Appendix A. Those different expression cassettes were introduced into pBIOS01957, a derivative of the pSB11 intermediate vector [41] harboring a GFP expression cassette under the control of the endosperm specific expressed High Molecular Weight Glutenin promoter and the *Streptomyces hygroscopicus* Bar gene under the control of the rice Actin1 promoter followed by its first intron [42]. The vectors were named, respectively, pBIOS02120, pBIOS02750, pBIOS02789 and pBIOS02792, and after homologous recombination with the acceptor vector pSB1 already present in *Agrobacterium tumefaciens* LBA4404 resulted in the super-binary vectors used for plant transformation. Maize transformation of the inbred line A188 with the *Agrobacterium tumefaciens* strain LBA4404 harboring a super-binary plasmid was performed, essentially as described by Ishida et al. [43] with the modifications described in [15]. Five to six independent transgenic events containing up to two inserted T-DNA copies were selected for the study. After an initial cross of the primary transformant (T_0_ plant) with the pollen of wild type (WT) A188 plants, two rounds of self-pollination were performed to obtain plants for which the cob carried only homozygous seeds. WT plants (line A188) were used as controls.

### 4.2. Plant Material and Growth Conditions

Maize (*Zea mays* L.) seeds from wild type (WT) and transgenic plants were first sown on coarse sand and after one week, when two to three leaves had emerged, individual plants were transferred to pots (diameter and height of 30 cm) containing clay loam soil with one plant per pot. Clay loam soil is composed of a mixture of loam (washed fine pit with no minerals) and loam balls of about 0.5 cm diameter that ensure sufficient aeration of the roots and allow growing the plant until maturity without lodging. Clay loam soil also allows a constant flow of the nutrient solution that is provided several times a day. Pots were placed in a glasshouse at the Institut National de la Recherche Agronomique (INRA), Versailles, France, and grown from May to September 2016. The glasshouse was equipped with an automatic cooling system (water mist + aeration) maintaining the temperature below 30 °C and humidity above 50%. Pots were moved every week to avoid shading and position effects. For each of the five to six transgenic events produced for each of the four different constructs and for the WT, four individual plants corresponding to four technical repetitions were grown until kernel harvest. These four individual plants correspond to the four replicates used for leaf metabolite profiling and yield-related traits measurements. For quantification of transcript abundance metabolic profiling at the 10 to 11 leaf stage, half of the eighth leaf was harvested for the vegetative stage (V) to obtain enough homogenous plant material representative of this plant developmental stage. For metabolic profiling, half of the leaf below the emerging ear, 15 days after silking (15DAS) was harvested. The leaf, below the ear, was selected since it has been shown to provide a good indication of the source sink transition during kernel filling [44]. Leaf samples were harvested between 9:00 and noon and frozen in liquid N_2_, ground to a homogenous powder, and stored at −80 °C. Harvesting of the ear (for yield-related traits measurements) and of the remaining shoot (including husks and cobs for shoot biomass measurements) was performed around 80 days after flowering. Kernel yield (KY), kernel number (KN) and thousand kernel weight (TKW) were determined according to the methods described in Bertin and Gallais [45]. In the glasshouse, plants were watered four times a day with 200 mL of a complete nutrient solution containing 10 mM N (8 mM NO_3_^−^ + 2 mM NH_4_^+^), [46]. The complete nutrient solution contained 5 mM K^+^, 3 mM Ca^2+^, 0.4 mM Mg^2+^, 1.1 mM H_2_PO_4_^−^, 1 mM SO_4_^2−^, 1.1 mM Cl^−^ 21.5 µM Fe^2+^ (Sequestrene; Ciba-Geigy, Basel, Switzerland), 23 µM B^3+^, 9 µM Mn^2+^, 0.3 µM Mo^2+^, 0.95 µM Cu^2+^ and 3.5 µM Zn^2+^ There was no major variation in the silking date between the WT and transgenic lines.

### 4.3. RNA Extraction, RT-Transcription and RT-qPCR Analysis

Total RNA was extracted as described by Verwoerd et al. [47] from frozen plant material stored at −80 °C. Quantification of total RNA was performed using a Nanodrop One instrument (Thermo Fisher, Illkirsh, France) and quality checked by measuring the A_260_/A_280_ ratio. Reverse transcription reactions and quantitative first strands were synthesized according to Daniel-Vedele and Caboche [48] using M-MLV reverse transcriptase and oligo(dT) 15 primers (Invitrogen, Carlsbad, CA, USA). The PCR reaction was performed on a Mastercycler^®^ ep *realplex* instrument (Eppendorf, Hamburg, Germany) with the Mesa Fast qPCR Mastermix Plus for SYBR Assay (Eurogentec, Liege, Belgium) according to the manufacturer’s protocol. Each reaction was performed on a 1:10 dilution of the first cDNA strand synthesized as described above, in a total reaction of 20 µL. With this dilution, the SYBR green signal was linear. The results were normalized by employing the transcript accumulation of the elongation factor 1α gene (*EF1**α*; EU960821) as constitutive reference using (2^−^^Δ^^Ct^)100. *EF1**α* was chosen as the reference gene from a set of 21 genes selected from a microarray experiment, which was further validated by RT-qPCRs using cDNAs from different organs and growth conditions. Each pair of primers was previously used in a RT-PCR to confirm the gene specificity amplification and their products were sequenced. The reaction efficiency of primer pairs was also calculated using an amplification standard curve. The slope of the standard curve gave the efficiency of the PCR reaction by the following equation: Efficiency = 10^(−1/slope)^ − 1. The sequences of the primers for the genes encoding NADH-GOGAT, NAD-ICDH, GDH1 and GS1.3 used in RT-qPCR are presented in Appendix A.

### 4.4. Enzyme Activities Measurements

Proteins were extracted from frozen leaf material stored at −80 °C. All extractions were performed at 4 °C. NADH-GOGAT and NAD-IDH activities were measured according to the protocol described by Singh and Srivastava [49] and Lemaître et al. [20] respectively. Glutamate dehydrogenase [NAD(H)-GDH] aminating and deaminating activities were measured as described by Turano et al. [50] except that the extraction buffer was that used by Tercé-Laforgue et al. [51]. Glutamine synthetase (GS) was measured according to the method of O’Neal and Joy [52] using the synthetase reaction.

### 4.5. Metabolite Extraction and Analyses

Frozen leaf tissues were reduced to a homogenous powder and stored at −80 °C until required for metabolite measurements. For the leaf metabolome analyses, all steps were adapted from the original protocol [53], following the procedure described by Tcherkez et al. [54].

The ground dried samples (10 mg DW) were resuspended in 1 mL of frozen (−20 °C) Water:Acetonitrile:Isopropanol (2:3:3) containing Ribitol at 4 µg/mL and extracted for 10 min at 4 °C with shaking at 1500 rpm in an Eppendorf Thermomixer. Insoluble material was removed by centrifugation at 14,000 rpm for 10 min. An amount of 100 µL was collected and 10 µL of myristic acid d27 at 30 µg/mL was added as an internal standard for retention time locking. Extracts were dried for 4 h at 35 °C in a Speed-Vac and stored at −80 °C. Samples were taken out of −80 °C, warmed 15 min before opening and Speed-Vac dried again for 1.5 h at 35 °C before adding 10 µL of 20 mg/mL methoxyamine in pyridine to the samples and the reaction was performed for 90 min at 30 °C under continuous shaking in an Eppendorf thermomixer. An amount of 90 µL of N-methyl-N-trimethylsilyl-trifluoroacetamide (MSTFA; REGIS TECHNOLOGIES Inc., Grove, IL, USA) was then added and the reaction continued for 30 min at 37 °C. After cooling, 100 µL was transferred to an Agilent vial for injection. Four hours after derivatization, 1 µL of the sample was injected in splitless mode on an Agilent 7890B gas chromatograph coupled to an Agilent 5977A mass spectrometer. The column was an Rxi-5SilMS from Restek (30 m with 10 m integra-guard column). An injection in split mode with a ratio of 1:30 was systematically performed for saturated compounds quantification. Oven temperature ramp was 60 °C for 1 min then 10 °C/min to 325 °C for 10 min. Helium constant flow was 1.1 mL/min. Temperatures were the following: injector: 250 °C, transfer line: 290 °C, source: 230 °C and quadripole 150 °C. The quadrupole mass spectrometer was switched on after a 5.90 min solvent delay time scanning from 50 to 600 u. Absolute retention times were locked to the internal standard d27-myristic acid using the RTL system provided in Agilent’s Masshunter software. Retention time locking reduces run-to-run retention time variation. Samples were randomized. A fatty acid methyl ester mix (C8, C9, C10, C12, C14, C16, C18, C20, C22, C24, C26, C28, C30) was injected in the middle of the queue for external RI calibration. Raw Agilent data files were analyzed with AMDIS (https://chemdata.nist.gov/dokuwiki/doku.php?id=chemdata:amdis). The Agilent Fiehn GC/MS Metabolomics RTL Library (version June 2008) was employed for metabolite identifications. Peak areas determined with the Masshunter Quantitative Analysis (Agilent) in splitless and split 30 modes. Resulting areas were compiled in one single Excel File for comparison. Peak areas were normalized to Ribitol (internal standard) and Dry Weight. Metabolite contents are expressed in arbitrary units (semi-quantitative determination). The metabolite levels were analysed using the online server MetaboAnalyst 4.0 based on the R package MetaboAnalystR [55]. Data processing was conducted through quantile normalization followed by a generalized log transformation and mean centering. After data processing values for Sucrose were discarded because of the predominance of the metabolite and because there were no differences between the leaf samples of the WT and of the transgenic plants. MetaboAnalyst 4.0 was used to perform a Hierarchical Clustering Analysis (HCA) using the Euclidean distance and the Ward clustering algorithm as parameters for the leaf metabolites detected at the V stage and 15DAS, PCA and *t*-test between the WT (A188) and the four independent transgenic events selected in each of the four different transgenic lines T02291, T02289, T0312 and T022308. Pearson correlations were made using *cor.test* function in R. Correlation heatmap was made using the R package *pheatmap* (https://github.com/raivokolde/pheatmap). Network visualizations of metabolite correlations in relation to yield-related traits and shoot biomass accumulation were calculated and visualized using Cytoscape v.3.7.1 [56].

### 4.6. Statistics

For enzyme activity measurements and yield-related traits measurements, the results are presented as mean values for four individual plants with standard errors (SE = SD/√(*n* − 1)). Statistical analyses for metabolomic data were performed using the MetaboAnalyst 4.0 [56]. Statistical analyses for shoot biomass, yield-related traits, and enzyme activities measurements were performed using Prism 5 (Graphpad, CA, USA). Significant differences (*p* ≤ 0.05) were considered between the transgenic and WT plants.

## 5. Conclusions

The overexpression of NADH-GOGAT, one of the key enzymes involved in the synthesis of glutamate, appeared not to be beneficial in terms of plant productivity, even when enzymes in charge of the supply of its substrate, 2-Oxoglutarate, were overexpressed at the same time. Nevertheless, the study provides new information on the regulation of the C/N balance during the synthesis of glutamate and derived amino acids. As already demonstrated when deciphering the role of the enzyme GDH [39], the control of glutamate homeostasis appears to be one of the major check points acting at the cross roads between the C and N assimilation pathways. Due to this tight control of glutamate synthesis and use, modifying the flux of C and N to enhance the production of derived amino acids appears to be more difficult than theoretically possible; even the known boosting enzyme (GS1) is overexpressed. Therefore, the use of gene pyramiding often proposed to increase NUE in plants needs to be more carefully designed by considering the importance of glutamate synthesis and turn-over and its impact on plant productivity.

## Figures and Tables

**Figure 1 plants-09-00130-f001:**
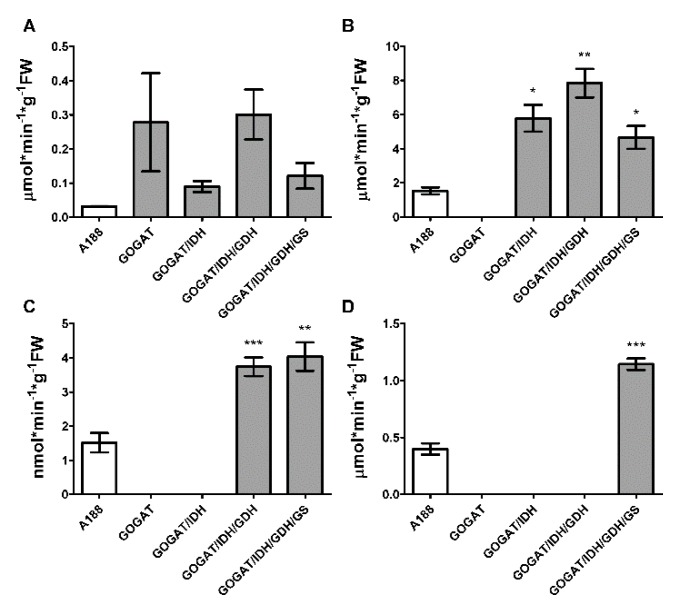
Enzyme activities of control and transgenic plants modified for 2-Oxoglutarate production and ammonium assimilation. The white column corresponds to the WT plants (A188). The grey columns correspond to transgenic line overexpressing NADH-GOGAT (construct T02291) and NADH-GOGAT stacked with NAD-IDH (construct T02289), NAD-IDH + NADH-GDH1 (construct T02308) and NAD-IDH + NADH-GDH1 + GS1.3 (construct T02312). NADH-GOGAT, (**A**). NAD-IDH, (**B**). NADH-GDH, (**C**). GS1.3, (**D**). The error bars correspond to the standard error (SE). Data were analyzed with a t-test. Significant differences are indicated with asterisks on top of the columns. * at *p* ≤ 0.05; ** at *p* ≤ 0.01, *** at *p* ≤ 0.001.

**Figure 2 plants-09-00130-f002:**
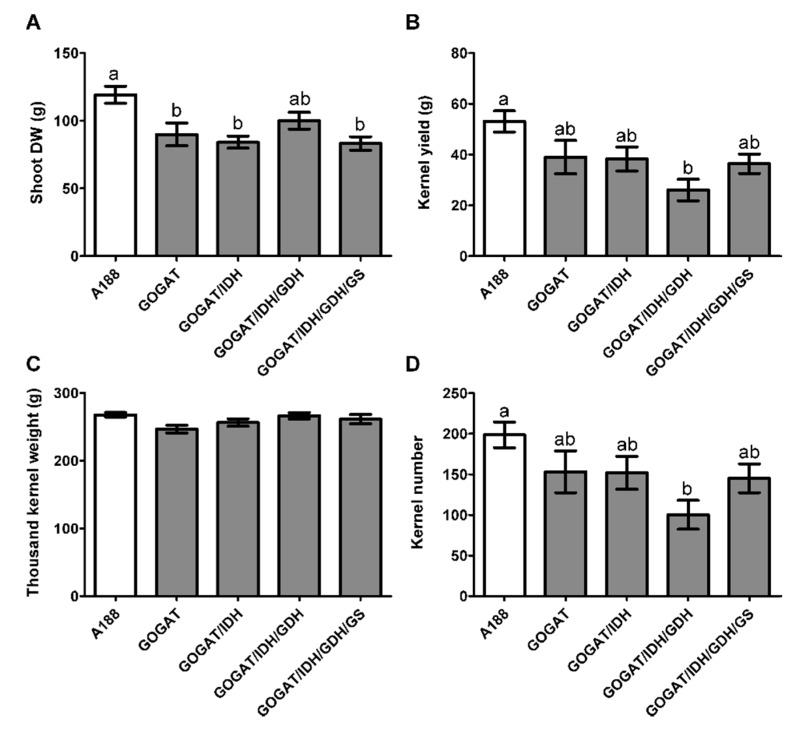
Shoot biomass production and yield-related traits in WT and transgenic plants modified for 2-Oxoglutarate production and ammonium assimilation. The white column corresponds to the WT plants (A188). The grey columns correspond to transgenic line overexpressing NADH-GOGAT (construct T02291) and NADH-GOGAT stacked with NAD-IDH (construct T02289), NAD-IDH + NADH-GDH1 (construct T02308) and NAD-IDH + NADH-GDH1 + GS1.3 (construct T02312). Shoot dry weight, (**A**). Kernel yield, (**B**). Thousand kernel weight, (**C**). Kernel number, (**D**). The error bars correspond to standard error (SE). Data were analyzed with a one-way ANOVA and the Newman-Keuls comparison test. The different letters (a, b) on top of the columns indicate significant differences at *p* ≤ 0.05.

**Figure 3 plants-09-00130-f003:**
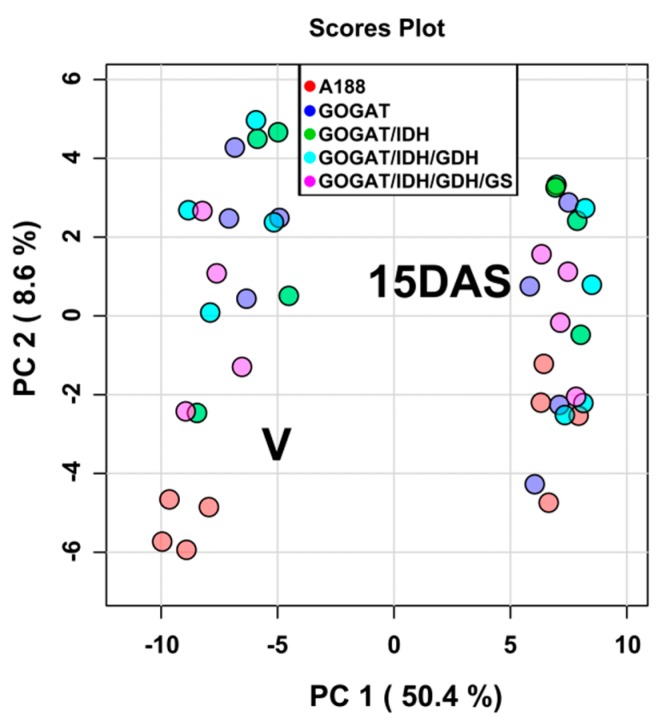
Principal component analysis (PCA) of the different leaf metabolites detected at the V stage and 15DAS of WT (A188) plants and transgenic maize lines overexpressing NADH-GOGAT (construct T02291) and NADH-GOGAT stacked with NAD-IDH (construct T02289), NAD-IDH + NADH-GDH1 (construct T02308) and NAD-IDH + NADH-GDH1 + GS1.3 (construct T02312). Metabolic and molecular traits were measured in the leaves of plants harvested at the vegetative (V) stage and 15 days after silking (15DAS). The four colored dots for each transgenic line correspond to the four independent transformation events selected for the metabolomic study. The values of the different metabolites were projected on to two biplots of principal components arranged in descending order of variance. Each component allowed the identification of groups of metabolites, depending on the plant developmental stage and of the nature of the line.

**Figure 4 plants-09-00130-f004:**
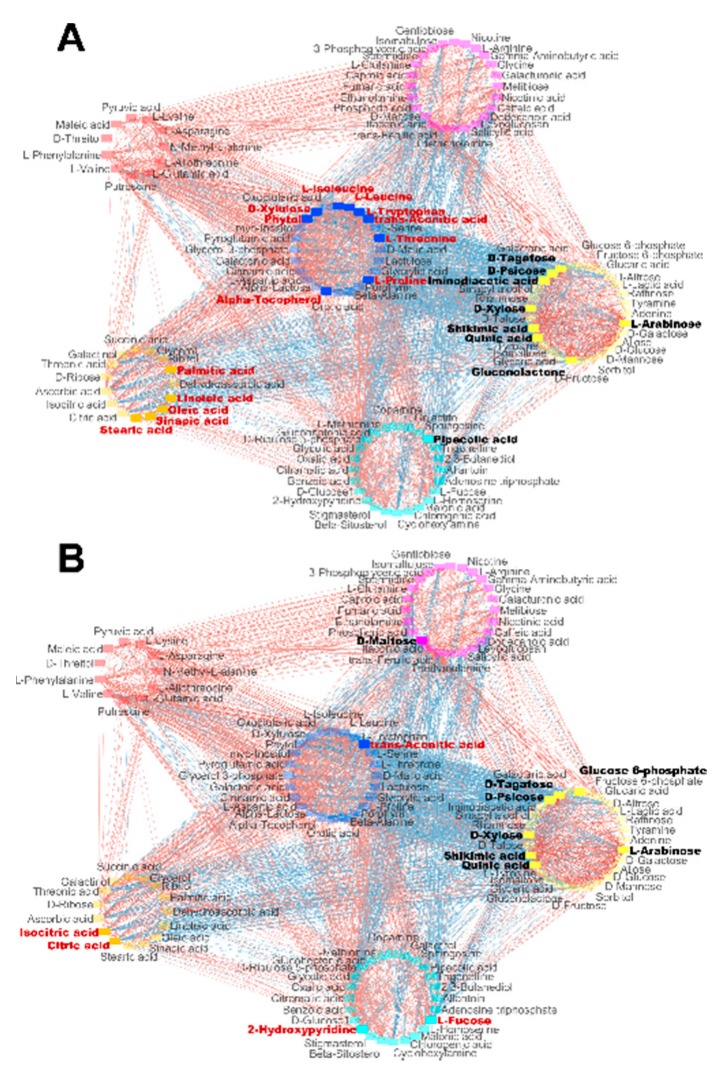
Network diagrams showing significant correlations (*p* ≤ 0.05) between leaf metabolite accumulation at the V stage and agronomic traits based on the calculation of Pearson coefficients. (**A**) Correlations between leaf metabolites and shoot dry weight (DW). (**B**) Correlations between leaf metabolites and kernel yield (KY). The red lines and blue lines represent positive and negative correlations between groups of metabolites, respectively. The names of the metabolites positively or negatively correlated with shoot DW and KY are highlighted in black and red, respectively. The circles correspond to the six groups of metabolites obtained following hierarchical clustering of the Pearson correlations between the amount of leaf metabolites at the V stage, shoot dry weight and kernel-related traits. The list of metabolites belonging to each of the six groups are provided in Appendix A.

**Figure 5 plants-09-00130-f005:**
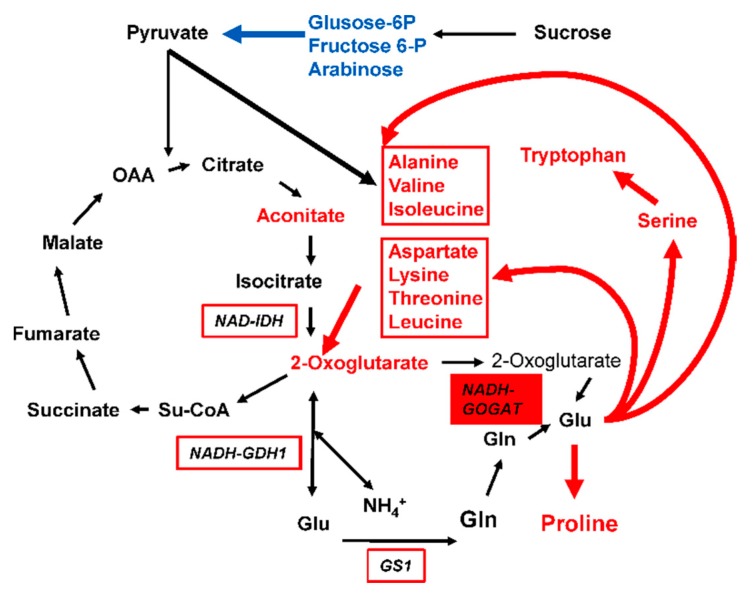
Scheme of amino acid biosynthesis derived from glutamate. Metabolites in red font are present in higher amounts in the leaves of maize plants overexpressing NADH-GOGAT (red box). Those in blue font are present in lower amounts. Red arrows indicate the putative metabolic pathways involved in the synthesis of amino acids and the release of 2-Oxoglutarate. The blue arrow corresponds to the use of carbohydrates to produce Pyruvate. The other enzymes used for gene pyramiding are boxed with a red line.

**Table 1 plants-09-00130-t001:** Metabolites exhibiting significant changes (*p* ≤ 0.05) between the WT (A188) and the different transgenic lines overexpressing NADH-GOGAT (construct T02291) and stacked with NAD-IDH (construct T02289), NAD-IDH + NADH-GDH1 (construct T02308) and NAD-IDH + NADH-GDH1 + GS1.3 (construct T02312). Results are expressed in Log of the Fold Change.

Trangenic Line	T02291	T 02289	T 02308	T 02312
Gene Overexpression	GOGAT	GOGAT/IDH	GOGAT/IDH	GOGAT/IDH/GDH/GS
2-Oxoglutarate	1.31	1.23		
L-Arabinose	−1.30	−1.21	−1.12	−1.14
D-Xylose	−1.00	−0.95	−1.01	−0.98
D-Tagatose	−1.78	−1.79	−1.72	−1.40
D-Psicose	−1.75	−1.83	−1.69	−1.40
Glucose-6-P	−0.72	−0.75	−0.59	
Fructose-6-P	−0.77	−0.83	−0.55	
L-Leucine	2.17	1.89	2.29	1.64
L-Valine	1.24	1.06	1.24	0.75
L-Serine	1.99	1.97	2.29	
L-Threonine	0.98	1.31	1.39	
β-Alanine	1.47	2.14	2.22	
L-Isoleucine	1.68	1.23	1.71	
L-Proline	1.52	1.43		
L-Tryptophan	1.42	1.58		
L-Aspartate	1.11			
L-Lysine	0.94

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
