# Peer review of "NADH-GOGAT Overexpression Does Not Improve Maize (Zea mays L.) Performance Even When Pyramiding with NAD-IDH, GDH and GS"

_plants, 2020, doi:10.3390/plants9020130_

Round 1
Reviewer 1 Report
In the manuscript by Cañas et al. it is checked whether pyramiding gene related to ammonium assimilation improves agronomic traits in maize. This work is interesting and can be found by a large group of recipients. The research objective is in the scope of Plants. The Authors perfectly introduce in their research area, as well as the results are well discussed. However, I have some of the following minor comments:
In the text of Abstract general information about NADH-GOGAT should be included.
Page 13, lines 421-433 - Instructions for Authors have been mistakenly included in the manuscript text - it should be removed.
Plant material and growth conditions
How many plants (total number) were used for research since it was only one experiment (May-September 2016)? These four individual plants correspond to the four replicates used for leaf metabolite profiling and yield-related traits measurements - I understand that these were technical repetitions?
What was the temperature during the experiment? Were there big fluctuations or was it controlled/recorded? In summer, in greenhouses (with poor air circulation) there may be a problem with relatively high temperatures. Please comment it.
What volume of nutrient solution was fed to the soil?
Page 9 line 297 shikimic acid instead of Shikimic acid
Page 14, line 486, p. 15 line 514 -80°C instead of -80oC
Page 15, line 547 sucrose instead of Sucrose
Author Response
Please see attachement + response to the Editor comments for information

Reviewer 2 Report
The article authored by Cañas RA and collaborators entitled “Does pyrimiding gene for ammonium assimilation enhances maize (Zea mays L) productivity” describes a comprehensive study where the authors intend to improve maize NUE by overexpressing NADH-GOGAT enzyme alone or pyramiding its overexpression with other enzymes related to ammonium assimilation or 2-OG production. Although big efforts to generate different maize transgenic lines where dedicated, the reported overexpressions did not improve plant performance but, in general, negatively affected plant growth. In any case, the results are relevant and merit to be shared with the community.
Comments:
1) NADH-dependent GOGAT enzyme is functional mainly in heterotrophic tissues, while Fd-GOGAT is the important enzyme for GS-GOGAT cycle in leaves. Although the absence of positive impact of the overerexpression is clear. All the measurements were performed in the leaves and therefore it would be interesting to know what happens in the root, or at least to discuss this point. Do you think the results would have been different if Fd-GOGAT had been overexpressed? May be that ammonium assimilation is being promoted in the root and this is a trade-off for shoot biomass and yield?
2) It is quite intriguing why the authors did not chose always to work with the maize genes but for NADH-GOGAT a wheat gene and for IDH a sorghum gene were overexpressed, while for GDH and GS the genes were from maize. It would be interesting that the authors include in the manuscript the reasons underlying this decision and to discuss it. Besides, genebank codes for the overexpressed genes should be given in materials and methods and/or Table S1.
3) I suggest changing the title for a more informative one, even if it will probably be more boring. Could it be “NADH-GOGAT overexpression does not improve Zea mays performance even when pyramiding with NAD-IDH, GDH and GS”
4) The plants were grown with 10 mM N supply. For the future, I think it will be interesting to grow the plants in N-limiting conditions. It may be that in this case the overexpression of these genes entails a benefit for the plant.
5) As done in Table 1. I think the figures would be more comprehensive by replacing the codes for the plant lines by GOGAT; GOGAT/IDH; GOGAT/IDH/GDH; GOGAT/IDH/GDH/GS
6) IN L567-568 I suggest to change the sentence “…even when the synthesis of one of its substrate 2-Oxoglutarate was boosted at the same time” by “..even when enzymes in charge of the supply of its substrate, 2-OG, were overexpressed at the same time”.
7) In L93 “ammonium assimilatory” should be replaced by “ammonium assimilation”
8) L517 a space is missing in “10mg”
Author Response
Please see the attachement including the response to the Editor comments
